# Double-Outlet Left Ventricle: Case Series and Systematic Review of the Literature

**DOI:** 10.3390/diagnostics13203175

**Published:** 2023-10-11

**Authors:** Michele Lioncino, Giulio Calcagni, Fausto Badolato, Giovanni Antonelli, Benedetta Leonardi, Andrea de Zorzi, Aurelio Secinaro, Gianluca Brancaccio, Sonia Albanese, Adriano Carotti, Fabrizio Drago, Gabriele Rinelli

**Affiliations:** 1Pediatric Cardiology and Cardiac Arrhythmias and Syncope Unit, Bambino Gesù Children’s Hospital, IRCSS, 00146 Rome, Italy; giulio.calcagni@opbg.net (G.C.); giovanni.antonelli@opbg.net (G.A.); benedetta.leonardi@opbg.net (B.L.); andrea.dezorzi@opbg.net (A.d.Z.); fabrizio.drago@opbg.net (F.D.); gabriele.rinelli@opbg.net (G.R.); 2Advanced Cardiothoracic Imaging Unit, Bambino Gesù Children’s Hospital, IRCSS, 00165 Rome, Italy; aurelio.secinaro@opbg.net; 3Cardiac Surgery Unit, Department of Pediatric Cardiology and Cardiac Surgery, Bambino Gesù Children’s Hospital, IRCCS, 00165 Rome, Italy; gianluca.brancaccio@opbg.net (G.B.); sonia.albanese@opbg.net (S.A.); adriano.carotti@opbg.net (A.C.)

**Keywords:** double-outlet left ventricle, DOLV, double-outlet ventricles, pulmonary root translocation, Rastelli

## Abstract

Double-outlet left ventricle (DOLV) is an abnormal ventriculo-arterial connection characterized by the origin of both great arteries from the morphological left ventricle. The aim of our paper is to describe the morphological and imaging features of DOLV and to assess the prevalence of the associated malformations and their surgical outcomes. METHODS From 2011 to 2022, we retrospectively reviewed the electronic case records of patients diagnosed with DOLV at the Bambino Gesu Children’s Hospital. A systematic search was developed in MEDLINE, Web of Science, and EMBASE databases to identify reports assessing the morphology and outcomes of DOLV between 1975 and 2023. RESULTS: Over a median follow-up of 9.9 years (IQR 7.8–11.7 y), four cases of DOLV were identified at our institution. Two patients were diagnosed with (S,D,D) DOLV subaortic VSD and pulmonary stenosis (PS): one patient had (S,D,D) DOLV with doubly committed VSD and hypoplastic right ventricle, and another patient had (S,D,L) DOLV with subaortic VSD and PS (malposition type). Pulmonary stenosis was the most commonly associated lesion (75%). LITERATURE REVIEW: After systematic evaluation, a total of 12 reports fulfilled the eligibility criteria and were included in our analysis. PS or right ventricular outflow tract obstruction was the most commonly associated lesion (69%, 95% CI 62–76%). The most common locations of VSD were subaortic (pooled prevalence: 75%, 95% CI 68–81), subpulmonary (15%, 95% CI 10–21), and doubly committed (7%, 95% CI 4–12). The position of the great arteries showed that d-transposition of the aorta was present in 128 cases (59% 95% CI 42–74), and l-transposition was present in 77 cases (35%, 95% CI 29–43).

## 1. Introduction

Double-outlet left ventricle (DOLV) is an abnormal ventriculo-arterial connection characterized by the origin of both great arteries, or more than 50% of each arterial root, from the morphological left ventricle [1,2]. Since its first description by Paul et al. [1], the morphology, associated lesions, and surgical outcomes of DOLV have been reported to exhibit significant heterogeneity, mainly because of its exceedingly low prevalence [3]. 

Among the conotruncal abnormalities, DOLV has been considered an embryological conundrum, and no more than 200 cases have been described since 1975. Moreover, the morphological features of DOLV are poorly understood and there is a lack of systematic data regarding their relative frequency. 

As for the double outlet right ventricle, surgical strategies for DOLV are based on the location of the ventricular septal defect (VSD), the relative position of the great arteries, and the presence of associated abnormalities [4]. Complete intraventricular repair, right-ventricle-to-pulmonary artery conduit, and pulmonary root translocation (PRT) have been proposed over time; nevertheless, selection criteria for intervention and head-to-head comparisons between surgical techniques are lacking. 

Over the last decades, with the evolution of cardiac surgery, long-term outcomes and late sequelae of DOLV are largely unknown [4]. In this case series, we report four patients affected with DOLV and evaluated them at Bambino Gesù Children’s Hospital, with a particular focus on their long-term outcomes. Furthermore, we performed a literature review to describe the anatomic details and the prevalence of associated lesions among the existing reports of DOLV. 

## 2. Methods

### 2.1. Institutional Data

From 2011 to 2022, we retrospectively reviewed case records, intra-operatory reports, and follow-up data of patients with a diagnosis of DOLV at Bambino Gesù Children’s Hospital, Rome. Considering the retrospective nature of the analysis, the current study did not require the approval of the local Ethics Committee according to current legislation. Data were retrospectively analyzed in accordance with personal data protection policies.

DOLV was diagnosed according to the definition provided by the Congenital Heart Surgery Nomenclature and Database Project [3]. Details of segmental analysis, location of the VSD, and great arteries were classified according to the international nomenclature for CHD [5]. Data on patients diagnosed before 2011 were excluded from the analysis. Twelve-lead ECG, echocardiographic scans, intraoperative descriptions, and other imaging techniques (CT scan and cardiac MRI) were reviewed by two independent authors (ML and GR) when needed. Controversies were adjudicated by a third author (FD). During follow-up, clinical evaluation, 12-lead ECG, 24 h electrocardiographic monitoring, echocardiogram, and, for children older than 7 years, exercise stress testing, based on the clinician’s judgement, were performed at regular intervals. 

### 2.2. Literature Sources and Review Strategy

We performed a systematic review in accordance with the updated Preferred Reporting Items for Systematic Reviews and Meta-Analyses guidelines [6] (Appendix A). A literature search was conducted in MEDLINE, EMBASE, and Web of Science databases to identify original articles published between 1 January 1975 and 30 May 2022. Cross-sectional studies, cohort studies, and case series were included. Single case reports and narrative reviews were excluded from the results. No publication date or publication status restriction was imposed. The term “double outlet left ventricle” was adopted in the electronic database search strategy and the bibliographies of all the original articles were analyzed to check for additional articles. Duplicates and manuscripts for which text was not available in English were excluded. Two authors (ML and GR) independently reviewed the full studies and disagreements were discussed with a third author (FD).

### 2.3. Data Extraction and Quality of the Included Studies 

We abstracted data on study design, year of publication, number of participants, and morphological features (location of the VSD, relative position of the great arteries, and associated abnormalities). When available, the data extraction strategy included infundibular morphology, visceroatrial situs, surgical strategy, follow-up time, and outcomes. Data extraction forms were completed by two independent authors (GC and GA). A modified version of the Newcastle-Ottawa scale (NOS) was adopted to evaluate the quality of the selected reports (Appendix A) [7]. Items related to comparability and adjustment of the control population were not included due to the absence of a control arm in most of the studies included in our analysis. The risk of bias was considered low when the total score of the modified NOS was 5, and high when it was equal to 3 or less. 

### 2.4. Statistical Analysis 

Due to the rarity of the condition and the descriptive nature of this case series, a qualitative analysis was performed when appropriate. Continuous variables with normal distribution were reported as mean **±** standard deviation. Skewed variables were presented as median **±** interquartile range. Categorical variables were presented as proportions. The 95% CIs were approximated using a binomial distribution. All statistical analyses were performed using R Studio (version 4.2.2, The R Foundation, Wien, Austria) 

## 3. Results

### 3.1. Case Series 

Over a median follow-up of 9.9 years (IQR 7.8–11.7 y), 4 cases of DOLV were identified at our institution. 

Pulmonary stenosis (PS) was the most common associated lesion (3/4). One patient presented with right ventricular hypoplasia and underwent Fontan palliation. Two out of four patients underwent a Rastelli-type operation, and in one patient, an REV operation was performed. 

### 3.2. S,D,D Double-Outlet Left Ventricle with RV Hypoplasia

Patient 1 was diagnosed with (S,D,D), double-outlet left ventricle, and a hypoplastic right ventricle. An echocardiographic evaluation revealed situs solitus, D-loop, and severe hypoplasia of the right ventricle. A large, unrestrictive atrial septal defect was present. The aorta overrode a large, doubly committed, juxta-arterial VSD with an absence of infundibular septum. The great arteries were transposed and the aorta was on the right and slightly posterior to the pulmonary artery, which was completely committed to the left ventricle. A tenuous mitro-aortic discontinuity and a well-developed subpulmonary conus were present. Crossed pulmonary arteries were observed, and a subsequent CT scan showed two adjacent, side-by-side coronary ostia, located in the anterior-facing sinus, which gave origin to the left anterior descending (LAD) and the right coronary artery (RCA). The left circumflex artery (LCx) had a retroaortic course originating from the RCA (Figure 1A–C). Pulmonary banding was performed as initial palliation. At 6 months of age, the patient was scheduled for Damus–Kaye–Stansel (DKS), and bidirectional Glenn palliation and extracardiac Fontan palliation were completed at 2.9 years. Follow-up MRI showed mild pulmonary valve regurgitation and pulmonary flow discrepancy between the LPA and RPA (ratio 33% versus 67%), with normal ventricular volumes and preserved ventricular function (LV end-diastolic volume 78 mL/m^2^, LVEF 53%). To date (follow-up time 5.3 years), the patient is in good clinical condition and his arterial oxygen saturation is stable (SpO2 85%). 

### 3.3. S,D,D Double-Outlet Left Ventricle and Pulmonary Stenosis

Patient 2 was diagnosed with complex congenital heart disease at another tertiary center and initial palliation with a modified Blalock-Taussig shunt (B-T) was performed. On admission, the diagnosis of (S,D,D), DOLV with subaortic VSD, and PS was confirmed. At 6 months of age (weight, 10 kg), a Rastelli-like operation, closure of the interventricular communication, and placement of an RV-to-PA bovine jugular vein conduit (Contegra, Medtronic Inc., Minneapolis, MN, USA, 18 mm) were performed. Long-term follow-up MRI (follow-up time: 13.1 years) showed significant dilatation of aortic root (maximal diameter 48 mm) and ascending aorta (38 mm, 21 mm/m^2^) and moderate aortic regurgitation (regurgitant fraction 33%). Mild left ventricular systolic dysfunction (LVEF 50%) with increased LV volume (LV end-diastolic volume 123 mL/m^2^) was present. The RV-to-PA conduit showed signs of calcification, and there was a significant discrepancy in pulmonary artery flow (LPA to RPA 23% vs. 77%). No major arrhythmia was reported during the follow-up. 

Patient 3 was diagnosed at birth with (S,D,D), DOLV with a subaortic VSD, and PS. The infundibular septum was hypoplasic. The aorta arose on the right and was posterior to the pulmonary trunk. The pulmonary artery was in fibrous continuity with the mitral valve, and a subaortic conus was present. The right coronary artery and left descending anterior artery originated from a single ostium from the anterior facing sinus. LCx was not visualized in echocardiography. A subsequent CT scan revealed that the LCx originate from the posterior-facing sinus, with a retroaortic course and slit-like origin (Figure 2). After the initial B-T shunt palliation, a Rastelli-type operation was performed at 1 year of age (10 kg), the interventricular communication was closed, and a bicuspidized pulmonary homograft was sewn in the right ventricular outflow tract. LPA was enlarged with the interposition of a Dacron patch. After a 6 year follow-up, severe degenerative stenosis of the RV-to-PA conduit was present; nevertheless, percutaneous conduit dilatation was contraindicated due to coronary abnormality. LCA and RCA were, respectively, caudal to supravalvular and subvalvular segments of the RV-to-PA conduit. After multidisciplinary discussion, an aortic homograft (25 mm) was implanted. At the last follow-up (total f.u. time 10.2 years), the patient was in good clinical status and presented moderate right ventricular outflow tract obstruction (gradient from tricuspid regurgitation at least 50 mmHg, between 50% and 75% of systemic arterial pressure). No arrhythmias were detected during follow-up, except for low-burden premature ventricular beats with superior axis and rBBB morphology, present at rest and during recovery, and disappearing with maximal exercise. 

### 3.4. S,D,L Double-Outlet Left Ventricle (“Malposition Type”)

Patient 4 was transferred from another center because of severe cyanosis (SpO2 40%). On admission, S,D,L-DOLV with subaortic VSD and PS were diagnosed. The aorta was anterior and to the left of the pulmonary artery and overrode a large conoventricular VSD. The pulmonary valve annulus was hypoplastic, and posterior deviation of the infundibular septum was evident. The mitral valve showed fibrous continuity with the pulmonary artery, and a well-developed subaortic conus was detected in 2D echocardiography (Figure 3D,E). A balloon atrial septostomy was performed to increase blood mixing, and the baby was then discharged. REV (Reparation à l’ètage ventriculaire) was performed at 1 year of age. After a follow-up period of 8.6 years, the patient showed a good clinical status. Significant pulmonary regurgitation was present at cardiac follow-up MRI (regurgitant fraction: 49%) with a mild increase in right ventricular volumes (end-diastolic volume 103 mL/m^2^). During periodic Holter monitoring, no significant arrhythmia was identified, except for the finding of ectopic atrial rhythm, without significant symptoms and with adequate chronotropic response during exercise stress testing. 

### 3.5. Review of the Literature

Using our systematic research strategy, we scrutinized 96 records for inclusion criteria published between 1 January 1975 and 30 May 2023 (Figure 4). After systematic evaluation, a total of 12 reports fulfilled the eligibility criteria and were included in our study. During the literature review, 84 reports were excluded from the study: 78 single case reports, 2 duplicates, and 4 without an available full text in the English language. The morphological findings and surgical outcomes are summarized in Table 1. 

### 3.6. Prevalence of Pulmonary Stenosis/Right Ventricular Outflow Obstruction (PS/RVOTO)

The presence of PS/RVOTO was assessed in all the examined studies (n = 12, 217 patients). Van Praagh et al. [8] described 109 hearts with DOLV, among these 73 (66%) had PS/RVOTO. Luciani et al. [9] reported surgical outcomes of DOLV (n = 22, median age at repair 48 months [IQR 0.3–336]). In this study, a systematic review of the literature was conducted to assess the surgical results. As individual data were available, we excluded patients who were already evaluated in our analysis from other studies. PS was present in 17/22 patients (66%). A similar prevalence was reported in a study by Bharati et al. [10] (n = 45), where PS was present in 28 patients (62%). Among these, in 9/45 cases, PS was associated with tricuspid valve abnormalities, such as tricuspid atresia or stenosis. Imai Compton et al. [11] reported the anatomical features and surgical management of 19 patients referred to The Hospital for Sick Children between 1960 and 2008. RVOTO was present in 6/19 patients (31%). Notably, pulmonary atresia was present in 7/19 patients. Smaller studies reported a significantly higher prevalence of PS. Namely, all included patients presented with PS in the studies by Brandt et al. [12], DeLeon et al. [13], Ootaki et al. [14], McElhinney et al. [15], Raja et al. [16], Villani et al. [17] and Stegmann et al. [18]. Of interest, PS was not reported in the study by Subirana et al. [19], which included hearts with atrioventricular discordance. 

**Table 1 diagnostics-13-03175-t001:** Summary of the included studies.

First a Author	Year of Publication	Nr of Patients	Location of VSD (n = Patient Count)	Associated Lesions (n = Patient Count)	Infundibular Morphology	Great Arteries	Visceroabdominal Situs	Surgical Strategy	Follow Up	Adverse Outcomes
Bharati et al. [10]	1978	45	VSD: subaortic (32), sub-pulmonary (8), double committed (4). IVS (1)	Dextrocardia (2); Pulmonary stenosis (28) Tricuspid Atresia (9), Tricuspid stenosis (5)	Aortic-mitral continuity Hypoplasic subpulmonary conus	D-aorta (23) L-aorta (19) Anterior aorta (3)	Solitus (45)	Not assessed	-	
DeLeon et al. [13]	1995	2	VSD: subaortic (2)	Pulmonary stenosis (2)	Not assessed	D-aorta (2)	Solitus (2)	Pulmonary root translocation (2)	Median follow-up 1.5 years (IQR 1.25–1.75)	Small residual VSD (1)
Raja et al. [16]	2020	4	VSD. Subaortic (4)	Pulmonary stenosis (3); Abnormal coronary arteries (3)	Not assessed	D-aorta (4)	Solitus (4)	Pulmonary root translocation (3) REV (1)	Median follow-up 13.5 months (11.5–15.5)	
Imai Compton et al. [11]	2010	19	Not assessed	ASD (14); RV hypoplasia (12) Tricuspid atresia (7) Pulmonary atresia (7) Pulmonary stenosis (6) LVOTO (5)	Not assessed	D-aorta (4) L-aorta (8) Not described (7)	Situs: not known	Univentricular (9) Biventricular (7) Other (palliative care, lost to follow up) (3)	Median follow-up 7.5 years (2 weeks-19 y)	5 deaths: heart failure (2), surgical related (1), aortic thrombosis (1), myocardial infarction (1) Residual lesions: aortic regurgitation (5), mitral regurgitation (2), arrhythmia (4, 1 PMK implantation); hypertension (2)
Ootaki et al. [14]	2001	2	VSD: subaortc (2)	Pulmonary stenosis (2)	Bilaterally absent	D-aorta (1) L-aorta (1)	Solitus (2)	Pulmonary root translocation (2) Pulmonary regurgitation	Follow-up 26 and 55 months	-
Subirana et al. [19]	1984	2	VSD: subaortic (2),	Hypoplastic RV (2), AV discordance (2); Pulmonary stenosis (1)	Bilaterally absent (2)	L-aorta (2)	Solitus (2)	Not assessed	-	-
Luciani et al. [9]	2017	22	VSD: subaortic (15), sub-pulmonary (3); double committed (2), NC (1), Not described (1)	Pulmonary stenosis (16), pulmonary atresia (1)	Not assessed	D-aorta (16) L-aorta (5) Not described (1)	Solitus (20) Situs inversus (2): 1 I,D,D 1 I,L,L	Extracardiac conduit (12) RVOT reconstruction (6) Intraventricular repair (3) Pulmonary root translocation (1)	Not evaluable	RV failure (4) Cerebral thrombosis (1) Atrioventricular block (1) Conduit change (2) Fontan (1)
Van Praagh et al. [8]	1989	109	VSD: subaortic (82), subpulmonary (16), Doubly-committed (7), Non-committed (3), IVS 1	PS/pulmonary stenosis (73) LVOTO (17)	Bilateral absence (20) Bilateral presence (8) Short subpulmonary (44) Subaortic (37)	D-aorta (70) L-aorta (38) A-aorta (1)	Solitus (103) Situs inversus (5): I,D,D (2) I,L,L (2) I,L,D (1) Right atrial isomerism (1) A,D,L (1)	Not assessed	-	
Brandt et al. [12]	1976	5	VSD: subaortic (5)	Pulmonary stenosis (5) Hypoplasic right ventricle (1)	Subpulmonary conus (2) Bilateral absence (2) Not assessed (1)	D-aorta (4) L-aorta (1)	Solitus (4) Situs inversus (1) 1 I,D,D	Extracardiac conduit (5) 2 late deaths	Median follow-up 2.25 years (1.5–2.5)	
McElhinney et al. [15]	1997	3	VSD: subaortic (3)	Pulmonary stenosis (2) Single coronary ostium (1)	Absent conus (2) Subaortic conus (2)	D-aorta (2) L-aorta (1)	Solitus (3)	Pulmonary root translocation (2) Extracardiac conduit (1)	Median fu: 39m (28–43)	
Villani et al. [17]	1979	2	VSD: subaortic (1), subpulmonary (1)	Pulmonary stenosis (2) Abnormal coronary arteries (1)	Not assessed	D-aorta (1) L-aorta (1)	Solitus (2)	Extracardiac conduit (2)	Follow up time NA	
Stegmann et al. [18]	1979	2	VSD: subaortic (1), subpulmonary (1)	Pulmonary stenosis (2)		D-aorta (1) L-aorta (1)	Solitus (2)	Extracardiac conduit (1) RVOT patch (1) and conduit	Follow up time NA	Reoperation (1) Right heart failure (2)

Among the studies included in our analysis, the pooled prevalence of PS/RVOTO was (69%, 95% CI 62–76%, Table 2). 

### 3.7. Location of Ventricular Septal Defect

The location of the interventricular communication relative to the great arteries was assessed in 198 patients. The location of the VSD was not accurately reported in the study by Imai-Compton et al. [11], and these patients were excluded from our analysis. Among the remaining 11 studies, a subaortic VSD was present in 149 patients (pooled prevalence 75%, CI 68–81%), and a subpulmonary VSD was present in 29 patients (pooled prevalence 15%, CI 10–21%). The absence of the infundibular septum in the context of a doubly committed, juxta-arterial VSD was identified in 13 patients (pooled prevalence 7%, 95% CI 4–12) (Table 3). 

A non-committed ventricular septal defect was present in four (1%) patients, and two (~0.5%) patients had an intact ventricular septum. In addition, anatomical data were not available for 1 out of 22 patients reported by Luciani et al. [9]. 

Among the VSD subtypes, the most common was subaortic perimembranous VSD, which was present in ~76% of the cases described by Van Praagh et al. [8]. A similar prevalence was reported in the studies by Bharati et al. [10] and Luciani et al. [9] (32/45, 71% and 15/22, 68%, respectively). On the other hand, among the studies by DeLeon et al. [13], Raja et al. [16], Ootaki et al. [14], Subirata et al. [19], Brandt et al. [12], and McEllhiney et al. [15], the VSD location was subaortic in virtually all the cases. 

### 3.8. Position of the Great Arteries and Segmental Anatomy

The relative position of the great arteries was assessed in all the included studies. Anatomic details were not assessed in seven patients according to the study by Imai-Compton et al. [11] and one of the cases reported by Luciani et al. [9]. The aorta arose on the right and slightly anterior or on the right and slightly posterior to the pulmonary trunk, being mostly side-by-side (D-aorta according to Van Praagh’s segmental anatomy classification) in 128 patients (pooled prevalence 59%, CI 95% 42–64). The aorta was anterior or side-by-side and to the left of the pulmonary artery (L-aorta) in 77 patients (pooled prevalence 35%, CI 95% 28–42) (Table 4). 

The aorta was directly anterior to the pulmonary trunk in four patients (1%). 

Segmental analysis was available for 209 out of 217 patients. Most of the patients were in situs solitus (n = 199, 95%). One patient in Van Praagh’s [8] series had visceroatrial situs ambiguous and heterotaxy syndrome with asplenia. A common atrium opened in the morphological left ventricle via a common atrioventricular valve. The right ventricle was rudimental and was represented by an infundibular outlet chamber not committed to any great vessel. Both great arteries were related to the left ventricle, and bilateral conus was present. 

Situs inversus was present in nine patients (4%): three patients had situs inversus, atrioventricular concordance, and L-aorta (I,L,L) [8,9]; four patients had situs inversus, atrioventricular discordance, and D-aorta (I,D,D) [8,9,12]; one patient had atrioventricular concordance in situs inversus, subaortic VSD, PS, and a well-developed subaortic infundibulum (I,L,D) [8].

### 3.9. Surgical Strategies and Outcomes

Surgical strategies were assessed in 8 out of 12 studies (n = 61) [9,11,13,14,15,16,17,18]. Details regarding surgical techniques were not available in the manuscript by Imai-Compton et al. [11], whose observations were limited to biventricular versus univentricular correction. Consequently, data regarding surgical technique were detailed for 42 patients. 

Extracardiac conduit placement was the most common surgical strategy (n = 21/42), followed by pulmonary root translocation reported in 10 patients, RVOT patch in 7, intraventricular repair in 3, and REV in 1 patient. 

PRT was reported in five studies [9,13,14,15,16] published after 1995. 

The follow-up time varied among different studies, with a median time of 21 months (IQR 15–35). Residual lesions included pulmonary regurgitation in two patients and VSD in one patient. 

Only one case of right ventricular failure was reported among patients who underwent PRT [9,20]; in this case, correction was performed at an earlier age (4 months), and RVOTO acutely developed after surgery. 

Among the patients who underwent other procedures (Rastelli-like with extracardiac conduit, REV, intracardiac repair), right heart failure was the most common postoperative adverse outcome (n = 6). One patient experienced cerebral embolism, and one, notably in situs inversus, showed transient complete atrioventricular block.

## 4. Discussion

Among complex CHD, DOLV is considered an embryological conundrum and its extreme rarity may be partially explained by the observation that it does not constitute a part of normal cardiovascular development [21]. During cardiac morphogenesis, the infundibulum has a central role in driving the physiological switch of the great arteries [22]. Bulbar rotation and partial reabsorption of the free wall of the subaortic infundibulum drive the aortic root on the left and posteriorly relative to the pulmonary trunk, resulting in fibrous continuity with the mitral valve. At the same time, the development of the subpulmonary infundibulum drives the pulmonary trunk superiorly and leftward. The intrauterine arrest of infundibular development leads the aortic root to be positioned on the same frontal plane and on the right relative to the pulmonary artery, thus leading to the Taussig–Bing anomaly [23,24]. If reabsorption of the subaortic and subpulmonary conus is complete, both semilunar valves result in fibrous continuity with the mitral valve, thus leading to complex CHDs such as transposition of the great arteries with the posterior aorta, or with further translocation of the pulmonary root, DOLV [1,2]. In this scenario, DOLV may be considered in the opposite spectrum of the Taussig–Bing malformation [24]. In fact, the absence of a subarterial conus in the Paul type of DOLV has been demonstrated in pathologic specimens by Van Praagh et al. [21]. In that specimen, the author described the presence of an atretic infundibulum positioned on the right sinus in complete dissociation with the great arteries. 

In our case series, three patients had pulmonary stenosis. This finding is concordant with the estimated prevalence of PS/RVOTO of ~69% (95% CI 62–75), according to our literature review. Interestingly, most cases with PS had subpulmonary conal musculature; nevertheless, this short and underdeveloped infundibulum is stenotic, similar to that found in the Tetralogy of Fallot. In our institutional experience, one case of DOLV had bilateral conus: this patient had right ventricular hypoplasia, and the finding of DOLV might be explained by ventricular underdevelopment rather than infundibular reabsorption. On the other hand, patient 4 showed malposition of the great arteries (L-aorta in D-loop ventricle) with mitro-pulmonary continuity and subaortic conus.

These cases might be explained by hypothesizing that some conditions, such as right ventricular hypoplasia and malposition of the great arteries (L-aorta from D-loop ventricle) may increase the chance of the great arteries being committed to the left ventricle irrespective of infundibular morphology [21]. 

Our study was the first to assess the long-term, contemporary outcomes of surgically corrected DOLV. After 9.9 years of mean follow-up, stenosis of the RV-to-PA conduit, aortic root dilatation, and aortic regurgitation were the most common late adverse outcomes among patients who underwent a Rastelli-like operation. In addition, none of the patients who underwent the Rastelli-like procedure developed left ventricular outflow tract obstruction. In contrast, pulmonary regurgitation and subsequent right ventricular dilatation have been described in the patients who underwent the REV procedure. 

Many surgical options have been proposed for DOLV. When pulmonary stenosis is not present, the choice of biventricular repair is influenced by the position of the ventricular septal defect and the presence of coronary artery abnormalities. In the case of subpulmonary, doubly committed or large subaortic VSD, baffling the right ventricle in the pulmonary trunk with a “boomerang patch” [25] or PRT has been reported. In our literature review, PRT was reported in five studies [9,13,14,15,16] published after 1995. Notably, the first patient who underwent this procedure developed right ventricular failure because of early surgery (4 months) [20]. The presence of subpulmonary conus represents a fundamental anatomic detail in patients scheduled for PRT as it facilitates the uprooting of the pulmonary artery on the right ventriculotomy. On the contrary, PRT seems less feasible when the pulmonary valve is in fibrous continuity with the aorta. 

In our case series, we reported a rare case of S,D,L DOLV who underwent REV operation. Notably, in this patient, a stenotic pulmonary valve was in fibrous continuity with the mitral valve, and thus PRT could not be performed. Recently, complete repair of the DOLV in infancy has been demonstrated to be safe and feasible [9]. In this scenario, REV may represent a surgical option, particularly among patients with pulmonary stenosis, where it may avoid the use of prosthetic conduits and does not require re-rooting of the native pulmonary valve. Further studies are needed to assess the best surgical management for biventricular repair of DOLV.

To date, our study provides the first example of multimodality imaging in DOLV. The integration of different imaging modalities may have a fundamental role in defining the patterns of coronary anatomy and surgical planning in this rare population.

## 5. Study Limitations

Due to the retrospective nature of our study, our results were limited by the presence of missing data. We could not identify patients treated before 2011 at our institution. Furthermore, anatomical details about infundibular morphology were not available for one patient.

Due to the large time interval included in our literature review, the rate of surgical adverse events has significantly changed over time; therefore, our results may not be generalizable to current surgical outcomes.

## 6. Conclusions

Among patients with DOLV, the prevalence of PS/RVOTO was (69%, 95% CI 62–76%). The most common locations of VSD were subaortic (pooled prevalence 75%, 95% CI 68–81), subpulmonary (15%, 95% CI 10–21), and doubly committed (7%, 95% CI 4–12). The position of the great arteries showed that d-transposition of the aorta was present in 128 cases (59% 95% CI 42–74), and l-transposition in 77 (35%, 95% CI 29–43).

## Figures and Tables

**Figure 1 diagnostics-13-03175-f001:**
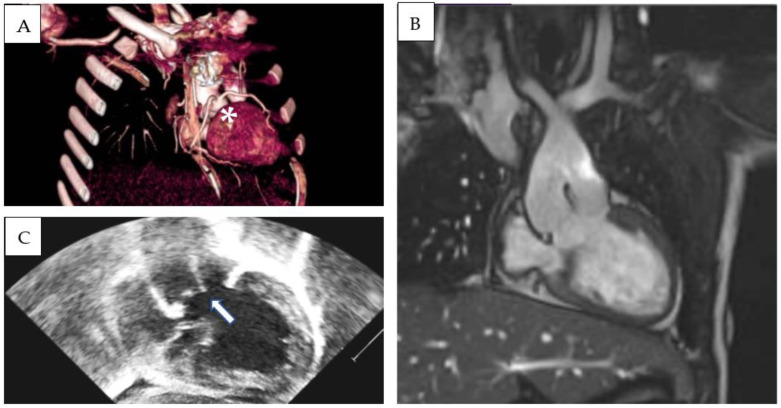
Three-dimensional CT-scan (**A**), MRI (**B**), and echocardiographic (**C**) subcostal view showing double-outlet left ventricle, subaortic VSD, d-transposed aorta and Damus–Kaye–Stansel palliation. Absence of an infundibular septum is a common finding (arrow). LAD and RCA originate from two adjacent, side-by-side coronary ostia from the anterior-facing sinus (asterisk).

**Figure 2 diagnostics-13-03175-f002:**
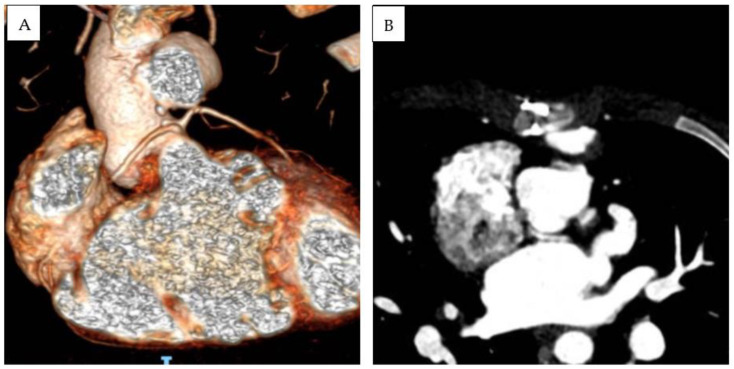
Three-dimensional (**A**) and MPR-angio CT scan (**B**) documenting anomalous coronary anatomy in Patient 3. Due to the strict relationship between RV-to-PA conduit and RCA and LCA, percutaneous conduit dilatation was contraindicated.

**Figure 3 diagnostics-13-03175-f003:**
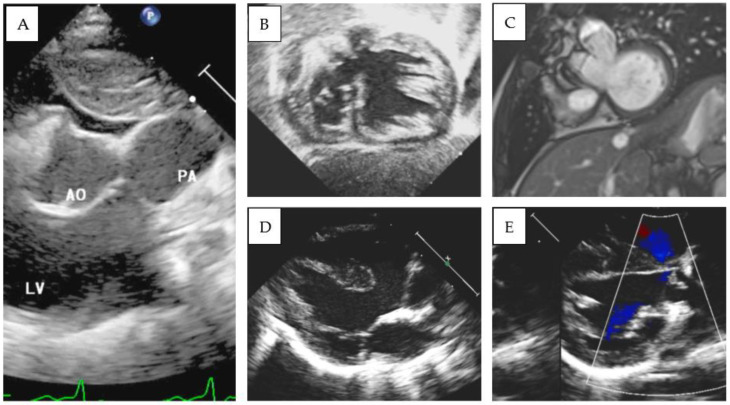
(**A**) 2D Echocardiographic parasternal short-axis view documenting both great arteries originating from the morphological left ventricle. (**B**) Echocardiographic and (**C**) MRI views documenting DOLV and hypoplasic right ventricle (patient 1). (**D**,**E**) Parasternal long-axis view documenting mitro-aortic discontinuity and mitro-pulmonary continuity in patient 4.

**Figure 4 diagnostics-13-03175-f004:**
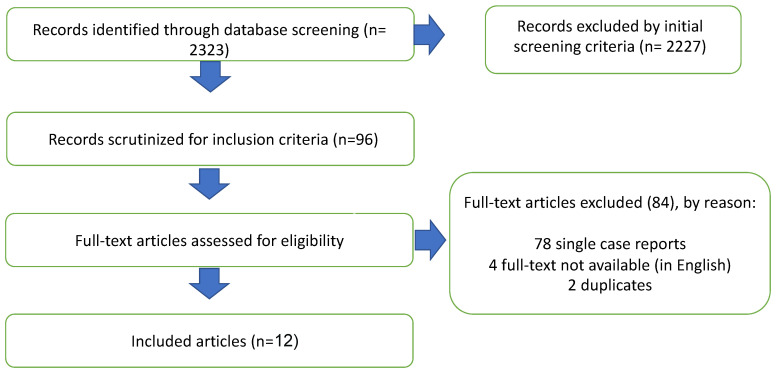
Flow chart of the study protocol.

**Table 2 diagnostics-13-03175-t002:** Prevalence of PS/RVOTO among patients diagnosed with DOLV.

Study	N. of Patients	PS/RVOTO	Prevalence (%)	95% Confidence Interval
Bharati	45	28	62	(46–76)
Brandt	5	5	100	(47–100)
De Leon	2	2	100	(15–100)
Imai Compton	19	13	68	(43–97)
Luciani	22	17	77	(54–92)
Subirana	2	1	50	(12–98)
Ootaki	2	2	100	(15–100)
Raja	4	3	75	(19–99)
Van Praagh	109	73	66	(57–75)
McElhinney	3	2	66	(9–99)
Villani	2	2	100	(15–100)
Stegmann	2	2	100	(15–100)
Pooled	217	73	69	(62–76)

**Table 3 diagnostics-13-03175-t003:** Location of ventricular septal defect among the included studies.

**Study**	**N**	**Subaortic** **VSD**	**Prevalence (** **95% CI)**	**Subpulmonary** **VSD**	**Prevalence** **(95% CI)**	**Doubly Committed VSD**	**Prevalence (** **95% CI)**
Bharati	45	32	71 (56–84)	8	18 (8–32)	4	9 (2–21)
Brandt	5	5	100 (48–100)	-	-	-	-
De Leon	2	2	100 (15–100)	-	-	-	-
Luciani	22	15	68 (45–86)	3	14 (3–35)	-	-
Subirana	2	2	100 (15–100)	-	-	-	-
Ootaki	2	2	100 (15–100)	-	-	-	-
Raja	4	4	100 (40–100)	-	-	-	-
Van Praagh	109	82	75 (66–83)	16	15 (9–23)	7	6 (3–13)
McElhinney	3	3	100 (29–100)	-	-	-	-
Villani	2	1	50 (1–99)	1	50 (1–99)	-	-
Stegmann	2	1	50 (1–99)	1	50 (1–99)	-	-
**Pooled**	**198**	**149**	**75 (68–81)**	**29**	**15 (10–21)**	**13**	**7 (4–12)**

**Table 4 diagnostics-13-03175-t004:** Relative position of the great arteries among the included studies.

**Study**	**Number of pt**	**D-Aorta**	**Prevalence (** **95% CI)**	**L-Aorta**	**Prevalence** **(95% CI)**
Bharati	45	23	51 (36–66)	19	42 (28–58)
Brandt	5	4	80 (28–99)	1	20 (1–72)
De Leon	2	2	100 (15–100)	-	-
Luciani	22	16	73 (50–89)	5	23 (8–45)
Subirana	2	-	-	2	100 (16–100)
Ootaki	2	1	50 (1–99)	1	50 (1–99)
Raja	4	4	100 (40–100)	-	-
Van Praagh	109	70	64 (54–73)	38	35 (26–45)
McElhinney	3	2	67 (9–99)	1	33 (1–91)
Villani	2	1	50 (1–99)	1	50 (1–99)
Stegmann	2	1	50 (1–99)	1	50 (1–99)
Imai Compton	19	4	21 (6–46)	8	42 (20–67)
**Pooled**	**217**	**128**	**59 (42–74)**	**77**	**35 (29–43)**

## Data Availability

Additional data are available from the corresponding author upon reasonable request.

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
