# Peer review of "Double-Outlet Left Ventricle: Case Series and Systematic Review of the Literature"

_diagnostics, 2023, doi:10.3390/diagnostics13203175_

Round 1

Reviewer 1 Report

Despite the limitations of the study, which the authors correctly stated in the article (lack of some retrospective data, comparison with modern surgical options), the work is undoubtedly very valuable and interesting for both cardiologists, surgeons and also pediatricians. I recommend that this material be published, as it will certainly arouse interest among medical professionals and will also be cited in future scientific articles.

Author Response

We thank the Reviewer for his comment and kindness. We hope this may represent only a small  contribution and hope that, in future years, a raising scientific interest will address many unanswered questions in the field of congenital heart disease.

Reviewer 2 Report

This manuscript is focused on the double-outlet left ventricle, a rare disease. The authors presented 4 cases.

The title must be changed as this is a case series. The presentation of the cases is very good with a lot of details. 

After that, there is a literature review. The authors presented a statistical analysis which is similar with the analysis of a meta-analysis. In reality only two studies have high number of cases and all the other studies had 2-4 cases. This analysis does not contribute significantly in the manuscript and create confusion. Must be removed. The results must be presented in Tables.

The discussion is good. 

Author Response

This manuscript is focused on the double-outlet left ventricle, a rare disease. The authors presented 4 cases.The title must be changed as this is a case series. The presentation of the cases is very good with a lot of details.  

We thank the reviewer for his valuable comment. Double-outlet left ventricle is an extremely rare condition and, according to the existing literature, only few pediatric centers worldwide have reported more than 4 patients.  We acknowledge this may account for the small sample size in our Institutional experience. Therefore, we changed our manuscript title according to Reviewer’s suggestions.

After that, there is a literature review. The authors presented a statistical analysis which is similar with the analysis of a meta-analysis. In reality only two studies have high number of cases and all the other studies had 2-4 cases. This analysis does not contribute significantly in the manuscript and create confusion. Must be removed. The results must be presented in Tables. The discussion is good. 

We thank the Reviewer for his valuable comment.

We acknowledge that, due to the extremely low prevalence of this condition, most of the studies included in our analysis have small number of cases. When studies included have small size, the variance of the proportion becomes very small leading to an inappropriately large weight in meta-analysis. Among the most common strategies used to overcome this limitation, Freeman-Turkey double arcsin or logit transformation have been used. In our study, we overcame the problem of small sample size calculating logit transformation of the proportions, transforming prevalence  to  a  variable that  is  not constrained to the 0–1 range and has approximately normal distribution,  we then conducted our  meta-analysis  and transformed  the  estimate back to a proportion1. This method has been considered scientifically accepted for the analysis of rare and small sample size studies2

Another small-study effect which may threaten the validity of a meta-analysis is represented by publication bias. This assumption is based on the idea that large studies are more likely to be published, even in case of small significance of the results, whereas small studies have greater risk of non-significant findings, and only very large effect sizes may reach significance when the sample size is small3. Publication bias and other small-size effects have been investigated and detected by inspection of funnel-plot asymmetry. In our paper, we used Egger’s regression test to identify a possible funnel plot’s asymmetry. Among the results, Egger’s regression test revealed possible publication bias in reporting the pooled prevalence of pulmonary stenosis and doubly committed VSD, whereas all the other reporting were not affected by publication bias. Furthermore, when Egger’s regression test detected a possible publication bias, we applied Duval and Tweedie Trimm and Fill Method5 to calculate bias-corrected estimate of the effect-size and the results did not significantly differ from the primary analysis, confirming the consistence of the results. Furthermore, leave-one out meta-analysis plots demonstrated that our results were consistent even with the exclusion of larger studies.

Nevertheless, we acknowledge that small sample may represent a possible limitation and removed the meta-analysis, as required by the Reviewer, to improve manuscript’s clarity.

We reported our results in Tables 2-4 and removed the sections relative to meta-analytic estimators and publication bias. Furthermore, we removed funnel plots and leave-one out plots from the supplemental material.

We highlited the changes in the updated manuscript.

  1. Murad MH, Sultan S, Haffar S, Bazerbachi F. Methodological quality and synthesis of case series and case reports.BMJ Evid Based Med. 2018;23(2):60-63. doi:10.1136/bmjebm-2017-110853
  2. Zhou S, Shen C. Statistical concerns for meta-analysis of rare events and small sample sizes.Lancet Infect Dis. 2022;22(8):1111-1112. doi:10.1016/S1473-3099(22)00364-4
  3. Borenstein, Michael, Larry V Hedges, Julian PT Higgins, and Hannah R Rothstein. Introduction to Meta-Analysis. John Wiley & Sons
  4. Rücker G, Carpenter JR, Schwarzer G. Detecting and adjusting for small-study effects in meta-analysis.Biom J. 2011;53(2):351-368. doi:10.1002/bimj.201000151
  5. Duval S, Tweedie R. Trim and fill: A simple funnel-plot-based method of testing and adjusting for publication bias in meta-analysis. 2000;56(2):455-463. doi:10.1111/J.0006-341X.2000.00455.X

Round 2

Reviewer 2 Report

It was a pleasure to review the above manuscript again.

I am happy to see that the authors answered all my comments. 

I dont have any further comments.